# What Is the Prognostic Significance of Culture-Documented Breakthrough Invasive Pulmonary Aspergillosis in Patients with Hematological Malignancies? A Propensity Score-Adjusted Analysis

**DOI:** 10.3390/jof11090623

**Published:** 2025-08-26

**Authors:** Sung-Yeon Cho, Sebastian Wurster, Takahiro Matsuo, Ying Jiang, Jeffrey Tarrand, Dimitrios P. Kontoyiannis

**Affiliations:** 1Department of Infectious Diseases, Infection Control and Employee Health, The University of Texas M.D. Anderson Cancer Center, Houston, TX 77030, USA; cho.sy@catholic.ac.kr (S.-Y.C.); stwurster@mdanderson.org (S.W.); takahirom1226@gmail.com (T.M.); yijiang@mdanderson.org (Y.J.); 2Division of Infectious Diseases, Department of Internal Medicine, Vaccine Bio Research Institute, Catholic Hematology Hospital, Seoul St. Mary’s Hospital, College of Medicine, The Catholic University of Korea, Seoul 06591, Republic of Korea; 3Section of Clinical Microbiology and Virology, Division of Pathology and Laboratory Medicine, The University of Texas MD Anderson Cancer Center, Houston, TX 77030, USA; jtarrand@mdanderson.org

**Keywords:** antifungal agents, *Aspergillus*, breakthrough infections, invasive pulmonary aspergillosis, leukemia, neutropenia

## Abstract

Mold-active prophylaxis has reduced the incidence of invasive pulmonary aspergillosis (IPA) in patients with hematological malignancies (HMs), but breakthrough IPA (Bt-IPA) is increasingly encountered. Therefore, we studied determinants of Bt-IPA risk and its prognostic significance. We retrospectively reviewed culture-positive proven/probable IPA cases in HM patients at MD Anderson Cancer Center (2016–2021). Bt-IPA and non-Bt-IPA cases were compared to characterize risk factors, clinical presentation, and outcomes. Independent predictors of 42-day all-cause mortality were assessed using propensity score-adjusted Cox regression. Among 118 IPA cases, 50 (42.4%) were Bt-IPA. Bt-IPA was associated with acute leukemia/myelodysplastic syndrome, active HM, severe neutropenia (<100/mm^3^), and graft-versus-host diseases. Uncommon *Aspergillus* species (non-*fumigatus*, *flavus*, *terreus*, or *niger*) were more frequent in Bt-IPA than non-Bt-IPA (20.4% vs. 4.8%, *p* = 0.010). Forty-two-day mortality was higher in Bt-IPA (65.3% vs. 37.3%, *p* = 0.003), but Bt-IPA itself was not an independent predictor or mortality (*p* = 0.064), which was instead driven by neutropenia (*p* = 0.020) and hypoalbuminemia (*p* = 0.002). In conclusion, Bt-IPA accounted for nearly half of contemporary IPA cases and was linked to host-related risk factors and the recovery of uncommon *Aspergillus* species. Although not an independent prognostic predictor, Bt-IPA reflected poor host status. Thus, early diagnosis, immune enhancement strategies, and effective first-in-class antifungals may improve outcomes.

## 1. Introduction

Invasive aspergillosis (IA) has been a long-feared complication in patients with hematological malignancies (HMs) and constitutes an important cause of morbidity and mortality [1,2,3]. Invasive pulmonary aspergillosis (IPA) is the most common clinical presentation of IA and is caused predominantly by *Aspergillus fumigatus* [4]. Antifungal prophylaxis with mold-active triazoles has resulted in a notable decrease in mycologically documented IPA cases and survival benefit compared to fluconazole or itraconazole [5].

Nonetheless, a subset of HM patients still develop breakthrough (Bt-) invasive fungal infections (IFIs), most commonly IA [6]. Since these patients typically have significant comorbidities, persistent neutropenia, and relapsed or refractory HM, Bt-IFIs are thought to be associated with poor outcome [7,8,9].

Given the widespread use of mold-active antifungals for prophylaxis [10], we aimed to study whether Bt-IPA, frequently encountered in our HM patients [7], is independently associated with poor prognosis or merely a marker of poor host status. As patients with Bt-IPA might substantially differ from the ones with non-Bt-IPA in terms of underlying risk of mortality, we used propensity score-adjustment to identify predictors of 42-day all-cause mortality in IPA to uncover the independent prognostic contribution of Bt-IPA.

## 2. Materials and Methods

### 2.1. Study Design and Population

We retrospectively reviewed electronic medical records of all consecutive adult (≥18 years) patients with culture-positive proven/probable IPA and HM at the University of Texas MD Anderson Cancer Center (January 2016–December 2021). Patients with non-malignant hematological diseases (e.g., aplastic anemia), positive cultures adjudicated as contamination, or prior fungal pneumonia were excluded from the analysis (Figure 1A). We collected data on demographics, clinical presentation, treatment history, HM disease status, laboratory data including therapeutic drug monitoring (TDM) results, antifungal treatment, and 42-day mortality after the first *Aspergillus*-positive culture. Antifungal exposure during the 90 days prior to IPA diagnosis was reviewed to evaluate Bt- status and prior antifungal agents. Information regarding *Aspergillus* spp. identification was derived from a clinical microbiology laboratory, based on macroscopic and microscopic morphology, growth characteristics, and MALDI-TOF MS (where applicable). Molecular identification (e.g., sequencing) to detect cryptic *Aspergillus* species was not performed as it has not been part of routine laboratory practice.

### 2.2. Definitions

IPA was defined according to the revised European Organization for Research and Treatment of Cancer/Mycoses Study Group (EORTC/MSG) consensus criteria [11]. Culture-positive proven/probable IPA cases were included only if they met the EORTC/MSG criteria, and *Aspergillus* species were identified from clinically significant respiratory specimens, including sputum, tracheal aspirates, bronchoalveolar lavage (BAL) fluid, or lung biopsy [12]. The date of IPA diagnosis was defined as the day of microbiological confirmation (i.e., date of specimen collection for culture confirmation), with the concomitant presence of clinical findings (including CT imaging and symptoms) compatible with IPA. Bt-IPA was defined, according to the Mycoses Study Group Education and Research Consortium and European Confederation of Medical Mycology consensus definition, as IPA occurring during the exposure of at least sufficient duration to achieve steady-state levels, regardless of the antifungal use context (prophylactic, empiric, pre-emptive, or targeted) [13]. Neutropenia and severe neutropenia were defined as absolute neutrophil counts of <500 cells/mm^3^ and <100 cells/mm^3^, respectively [11]. Significant corticosteroid use was defined as a prednisolone-equivalent dose of ≥0.3 mg/kg for ≥3 weeks within the 60 days prior to IPA diagnosis [11]. TDM for voriconazole or posaconazole was commonly performed and included in the analysis if measured within ± 5 days of Bt-IPA diagnosis [14]. Underlying HMs were considered active if any of the following criteria were met: (i) ongoing chemotherapy (including newly diagnosed HMs), (ii) stable disease not in remission, or (iii) relapsed/refractory disease. Significant graft-versus-host disease (GvHD) included acute GvHD grade ≥ 2 and moderate to chronic GvHD [15]. Chronic kidney disease was defined as an estimated glomerular filtration rate <60 mL/min/1.73 m^2^ for ≥ 3 months [16]. Chronic liver disease included chronic hepatitis B or C, alcohol-related liver disease, or nonalcoholic fatty liver disease [17]. Chronic lung disease was defined as the presence of chronic obstructive pulmonary diseases, asthma, or bronchiectasis.

### 2.3. Statistical Analyses

For univariate analysis, categorical variables were compared using Chi-square or Fisher’s exact tests. Continuous variables were compared with Student’s *t*-test or Wilcoxon rank sum test, depending on data distribution. Survival curves were compiled using the Kaplan–Meier method and compared by the Mantel–Cox log-rank test. A Cox proportional hazards regression model was first applied to identify the independent risk factors for 42-day all-cause mortality after IPA diagnosis. Variables with *p*-values ≤0.20 in univariate analyses were included in the initial multivariate models. A backward elimination procedure was then applied, which iteratively removed variables with the highest *p*-values until only those with *p* < 0.05 remained in the final model. Bt-IPA, the main variable of interest, was retained in the final model for 42-day all-cause mortality regardless of its *p*-value.

To further investigate the impact of Bt-IPA on 42-day mortality with reduced bias, we performed a propensity score-adjusted Cox proportional hazards regression using inverse probability of treatment weighting (IPTW). Therefore, we first determined propensity scores and calculated inverse probability of treatment (IPT) weights using logistic regression models that included all baseline variables with a *p*-value ≤0.20 in their univariate association with Bt-IPA. Subsequently, an IPT-weighted Cox proportional hazard model was compiled to estimate the 42-day all-cause mortality risk for Bt-IPA versus non-Bt-IPA. This model additionally included independent predictors identified in the unadjusted model, variables with residual imbalance after weighting, and *Aspergillus* species as a potential effect modifier. All tests were 2-sided with a significance level of 0.05. Data were analyzed with SAS v9.4 (SAS Institute Inc., Cary, NC, USA) and Prism v10.0.3 (GraphPad Software).

## 3. Results

### 3.1. Patient Demographics

During the 6-year study period, 118 culture-positive proven/probable IPA cases were included in the analysis after applying the exclusion criteria (Figure 1A). Patient characteristics are summarized in Table 1.

The median age was 63 years (range 22–89), and 63.6% of the patients were male. The proportions of patients with acute leukemia (AL) or myelodysplastic syndrome (MDS) (47.5%) and those with lymphoma, multiple myeloma (MM), or chronic leukemias (52.5%) were similar. Most patients (79.7%) had active HM and were undergoing chemotherapy. Neutropenia and severe neutropenia at IPA diagnosis were present in 41.5% and 32.2% of patients, respectively.

### 3.2. Prior Antifungal Drug Exposure of Patients with Bt-IPA

Among the 118 IPA cases, 50 (42.4%) met the definition of Bt-IPA, after excluding patients who had received antifungal therapy for too short a duration to reach a steady state. Bt-IPA to triazoles was the most common (34/50, 68.0%; 14 voriconazole, 16 posaconazole, 7 isavuconazole, including 2 cases with ≥2 triazole exposures), followed by echinocandins (25/50, 50.0%; 19 caspofungin, 6 anidulafungin) and liposomal amphotericin B (10/50, 20.0%). Seventeen patients (34.0%) had prior exposure to multiple antifungals at Bt-IPA diagnosis (Figure 1B). The median duration of antifungal exposure before Bt-IPA was 32.5 days (interquartile range, 11–51) (Figure 1C). Given the widespread use of long-term azole prophylaxis in our high-risk patients, the median duration of azole exposure prior to Bt-IPA (37 days) was significantly longer than that of non-azole antifungals (11 days; *p* < 0.001).

### 3.3. Clinical Characteristics of Bt-IPA vs. Non-Bt-IPA

Compared to patients with non-Bt-IPA, the Bt-IPA group was significantly more likely to have active HM (90.0% vs. 72.1%, *p* = 0.017), neutropenia (58.0% vs. 29.4%, *p* = 0.002), severe neutropenia (52.0% vs. 17.7%, *p* < 0.001), prior allogeneic hematopoietic cell transplantation (alloHCT) (32.0% vs. 10.3%, *p* = 0.003), and GvHD (18.0% vs. 5.9%, *p* = 0.038). Bt-IPA patients were also significantly more likely to have AL/MDS as their underlying HM than non-Bt-IPA patients (68.0% vs. 32.4%, *p* < 0.001). Conversely, chronic lung disease was more common in non-Bt-IPA patients than in those with Bt-IPA (16.2% vs. 2.0%, *p* = 0.012). Additionally, antifungal treatment patterns differed significantly between Bt-IPA and non-Bt-IPA groups (*p* < 0.001); combination therapy was more frequently given in Bt-IPA patients, likely reflecting salvage or escalation strategies, while azole monotherapy was the predominant approach in non-Bt IPA (Table 1).

### 3.4. Distribution of Aspergillus Species

A total of 124 *Aspergillus* isolates were identified from 118 IPA patients. Excluding six cases reported as ‘*Aspergillus* spp.’ or ‘*Aspergillus* spp. other than *A. fumigatus’*, *A. fumigatus* was the most common species (n = 48, 40.7%), followed by *A. niger* (n = 22, 18.6%), *A. terreus* (n = 21, 17.8%), and *A. flavus* (n = 14, 11.9%). Uncommon *Aspergillus* species included *A. ustus/calidoustus* (n = 5)*, A. versicolor* (n = 5)*, A. nidulans* (n = 2), and *A. glaucus* (n = 1). The proportion of uncommon *Aspergillus* species other than the four most common sections (*Fumigati, Flavi, Terrei,* and *Nigri*) was significantly higher in the Bt-IPA group (20.4%) than in patients with non-Bt-IPA (4.8%; *p* = 0.010) (Table 1).

### 3.5. Therapeutic Drug Monitoring

Serum TDM results within 5 days of Bt-IPA diagnosis were available for most patients (20/30, 76.7%), including 10/14 (71.4%) exposed to voriconazole and 13/16 (81.3%) exposed to posaconazole. Ninety percent (9/10) of voriconazole TDM results were ≥1.0 µg/mL (therapeutic range, 1.0–5.5), with a median serum level of 2.75 µg/mL (range, 0.9–7.4) at the time of Bt-IPA diagnosis. Only one patient had a voriconazole level below the therapeutic range. For posaconazole, 30.8% (4/13) of Bt-IPA cases had sub-therapeutic levels (<0.7 µg/mL), and the median level was 1.09 µg/mL (range, 0.38–2.60) (Appendix B
Figure A1).

### 3.6. Univariate Analysis of Factors Associated with 42-Day Mortality After IPA Diagnosis

Forty-two-day all-cause mortality after IPA diagnosis in our cohort was 49.1%, and was significantly higher among patients with Bt-IPA (65.3%) compared to the non-Bt-IPA group (37.3%, *p* = 0.003) (Table 1). This observation was corroborated by Kaplan–Meier survival curve analysis (Log-rank test, *p* = 0.005) (Figure 2).

Compared to 42-day survivors, deceased patients were more likely to have active HM (89.5% vs. 69.5%, *p* = 0.008), neutropenia or severe neutropenia at IPA diagnosis (57.9% vs. 27.1%, *p* < 0.001; 47.4% vs. 18.6%, *p* = 0.001), hypoalbuminemia <3.0 g/dL (64.9% vs. 27.1%, *p* < 0.001), and Bt-IPA (56.1% vs. 28.8%, *p* = 0.003) (Table 2). Furthermore, neutropenia dynamics, especially neutropenia recovery within 42 days from IPA diagnosis, differed significantly between survivors and non-survivors (45.6% vs. 17.0%, *p* = 0.002). In contrast, there was no difference in the median duration of antifungal exposure before Bt-IPA diagnosis between survivors and non-survivors (35 days vs. 31 days, *p* = 0.602) (Table 2).

### 3.7. Independent Predictors of 42-Day Mortality After IPA Diagnosis

In multivariate Cox regression analysis, only neutropenia at IPA diagnosis (adjusted hazard ratio [aHR] 1.95; 95% CI 1.11–3.41, *p* = 0.020) and hypoalbuminemia (aHR 2.52; 95% CI 1.42–4.47, *p* = 0.002) independently predicted 42-day mortality (Table 3A). After IPTW adjustment, the baseline balance between Bt-IPA and non-Bt-IPA patients improved significantly, with a 42.2–92.4% reduction in standardized mean differences (Figure 3, Appendix B
Table A1). Achieving a balance between groups through IPTW is important for reducing confounding and improving the validity of the mortality analysis. In the resultant IPT-weighted Cox proportional regression, neutropenia (aHR 2.10; 95% CI 1.13–3.88, *p* = 0.018) remained as the only independent risk factor for 42-day mortality (Table 3B). In contrast, Bt-IPA itself was not independently associated with increased mortality after IPTW adjustment for confounding factors (aHR 1.49; 95% CI 0.81–2.72, *p* = 0.201), suggesting that excess mortality may be attributable to differences in underlying host characteristics or clinical status.

## 4. Discussion

Although Bt-IPA is common in high-risk HM patients in an era of mold-active antifungal prophylaxis, clinical data on its prognostic significance remain limited. Higher mortality rates for Bt-IFIs have been reported in studies involving heterogenous patient populations and fungal etiologies, including both yeast and mold infections [18]. Recognizing the challenges of evaluating breakthrough infections in mixed pathophysiological settings, we focused on culture-positive proven/probable IPA within a relatively large and homogeneous patient cohort. Using these stringent inclusion criteria, our data show that Bt-IPA accounts for a substantial proportion (42.4%) of IPA cases in HM patients.

We found that Bt-IPA was significantly more common in patients with active HM, severe neutropenia, prior alloHCT, and/or GvHD, which represent key clinical indications for antifungal prophylaxis but also scenarios of significant underlying immune compromise. It is therefore challenging to determine whether Bt-IPA is the mere result of a severely compromised host defense driving Bt-IPA. The role of immune failure in the reduced efficacy of antifungals has long been emphasized [19] and supported by prior reports of an increased risk of Bt-IPA during mold-active azole prophylaxis in leukemia patients with persistent neutropenia [7,14,20].

On univariate analysis, variables related to poor host status, including active malignancy and unresolved neutropenia within 42 days from IPA diagnosis, were significantly associated with increased mortality. This observation aligns with a post-hoc analysis of the SECURE study, which showed suboptimal antifungal efficacy in patients with persistent neutropenia [21]. Similarly, multivariate Cox regression analysis demonstrated that only host factors, particularly neutropenia and hypoalbuminemia, were independently associated with 42-day mortality. A key strength of this study was the use of IPTW, allowing for propensity score adjustment without patient exclusion, a common disadvantage of propensity score matching. Thereby, IPTW allowed for a robust analysis of the impact of Bt-IPA in this high-risk HM patient population with greatly reduced bias. After IPTW adjustment, which significantly improved the baseline balance between Bt-IPA and non-Bt-IPA patients, neutropenia at the time of IPA diagnosis remained the sole factor independently associated with 42-day mortality. In contrast, neither the baseline nor IPTW-adjusted regression model identified a significant association between mortality and Bt-IPA per se.

While hypoalbuminemia could reflect disease-related inflammation, it is also a known prognostic marker for poor outcomes in bacterial and fungal infections [22]. Furthermore, albumin can neutralize toxins and modulate *Candida* pathogenicity [23]. Albumin also selectively inhibits the growth of *Mucorales* in vitro [24]. Thus, albumin-deficient mice were more susceptible to mucormycosis, and severe hypoalbuminemia was associated with poor prognosis in patients with mucormycosis [24]. Additionally, hypoalbuminemia alters the proportion of unbound azoles, affecting pharmacokinetics in critically ill patients, potentially leading to adverse outcomes [25]. However, due to the small sample size, no significant relationship was found between hypoalbuminemia and TDM levels in this study.

Interestingly, patients with lymphoma, MM, or chronic leukemias accounted for 52.5% of IPA cases in our cohort overall, suggesting that these diseases, rather than traditional AL/MDS, have become dominant for IPA. This may reflect undesirable immune effects of the monoclonal antibodies or newer immunomodulatory agents (e.g., alemtuzumab, ibrutinib, pomalidomide, or chimeric antigen receptor T-cell therapy, etc.) in these patients, where mold-active prophylaxis is not routinely used [26,27]. Thus, the benefit of antifungal prophylaxis for patients with individual risk factors, such as novel anti-cancer drugs or cell therapy, warrants further study.

Restricting our analysis to culture-positive proven/probable Bt-IPAs enabled us to assess the distribution of causative *Aspergillus* species and exclude galactomannan-positive *Hyalohyphomycetes* such as *Fusarium* [28]. We found that uncommon *Aspergillus* species accounted for more than one fifth of Bt-IPA cases, a proportion that was significantly greater than in the non-Bt-IPA group. This aligns with a retrospective study from Duke University reporting *A. ustus* in 43% of Bt-IPA cases under azole prophylaxis [29]. Consistent with another small case series [30], we identified a high proportion (57%) of *A. niger* isolates in patients with Bt-IPA to isavuconazole. However, larger datasets and additional data, such as antifungal susceptibility profiles and systematic cryptic species-level identification, are required to validate associations between individual *Aspergillus* species and breakthrough infections related to specific antifungal agents. For instance, further investigations regarding the role of cryptic species with intrinsic azole resistance (e.g., *A. lentulus* in section *Fumigati* or *A. tubingensis* in section *Nigri*) in Bt-IPA would be needed to guide clinical management.

The association between serum voriconazole or posaconazole levels and Bt-IFIs incidence or outcome remains controversial [6,9,31,32]. Consistent with a recent systematic review [6], low serum levels accounted for a minority of Bt-IPA cases in our cohort (10.0% of voriconazole-Bt-IPA and 30.8% of posaconazole-Bt-IPA cases). Of note, all posaconazole-Bt-IPA patients had received tablets. The relatively low proportion of sub-therapeutic triazole levels in our Bt-IPA cohort supports the importance of HM remission and host immune status for IPA outcomes, as opposed to antifungal drug levels or the duration of antifungal exposures [6,7,8,14].

This study has several limitations. Firstly, our study was not designed to assess the incidence or risk of Bt-IPA among all patients receiving antifungal prophylaxis. Given that the presence of IPA was a key inclusion criterion, the study lacked the denominator necessary to accurately evaluate the risk factors for developing breakthrough infection. Secondly, due to our focus on culture-documented proven/probable IPAs, our cohort might have been biased toward more severe cases where molds grew despite prior triazole exposure in AL or GvHD patients, as well as less susceptible hosts (e.g., lymphoma, MM) without prior mold-active prophylaxis. Thirdly, we did not assess antifungal resistance or cryptic species identification, as these are not routinely performed in our microbiology laboratory, and data for the in vitro/in vivo correlation of antifungal susceptibilities and clinical outcomes of IA are still lacking [33]. Furthermore, the single-center nature of this study limits its generalizability to other institutions or host populations. Specifically, the incidence of IPA, including Bt-IPA, may vary across centers, influenced by the proportion of high-risk HM patients, diagnostic strategies, and antifungal use. Finally, due to the retrospective study design and the inability to assess the infecting inoculum in IPA, we did not provide information on preexisting occupational exposures and habits that could be linked to high *Aspergillus* inoculum exposures and, consequently, higher risk for Bt-IPA [34,35].

In conclusion, our retrospective study at a tertiary cancer center revealed that Bt-IPA was common among patients with culture-positive IPA, particularly those with AL/MDS, active HM, neutropenia, and GvHD. Uncommon *Aspergillus* species accounted for more than one fifth of Bt-IPA cases. Although 42-day mortality was higher in patients with Bt-IPA, only host-related factors, especially neutropenia, were independently associated with 42-day mortality. Given that Bt-IPA reflects poor host status, earlier diagnosis, immune-enhancing strategies, and effective first-in-class antifungals could improve outcomes in Bt-IPA patients.

## Figures and Tables

**Figure 1 jof-11-00623-f001:**
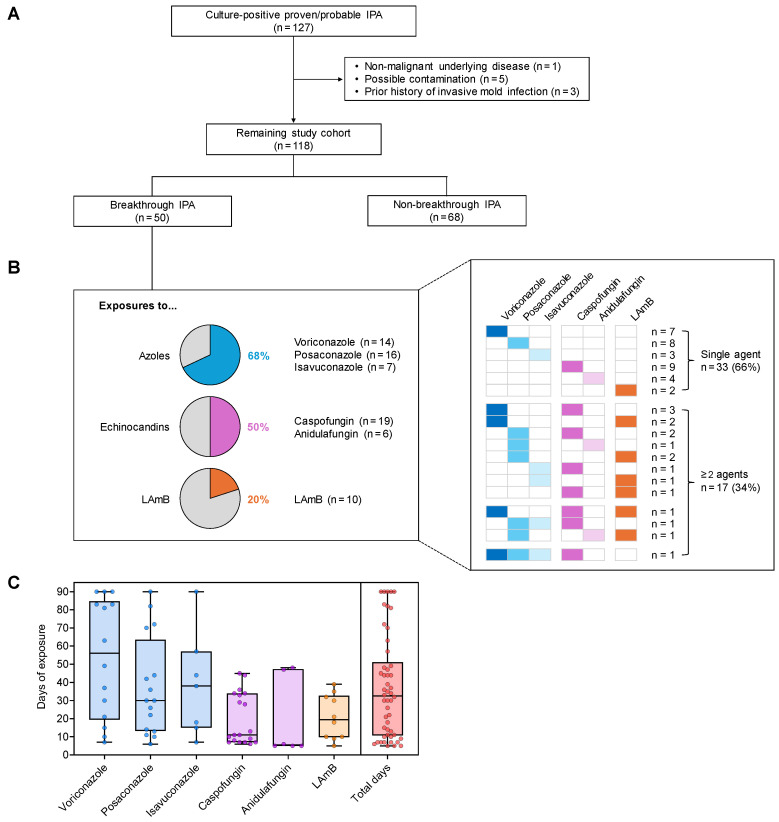
Breakthrough invasive pulmonary aspergillosis (Bt-IPA) is common in patients with hematologic malignancy and is associated with poor outcomes. (**A**) Flow chart summarizing the study population. (**B**) Summary of antifungal exposures at or before the diagnosis of Bt-IPA. Seventeen patients were exposed to more than one antifungal either concomitantly or sequentially. (**C**) Durations of antifungal exposure prior to Bt-IPA for each antifungal agent and on aggregate. Individual data points, (boxes), and ranges (whiskers) are shown. (**B**,**C**) Exposures of significant duration to reach steady-state levels were considered within a period of 90 days prior to Bt-IPA. Abbreviation: LAmB, liposomal amphotericin B.

**Figure 2 jof-11-00623-f002:**
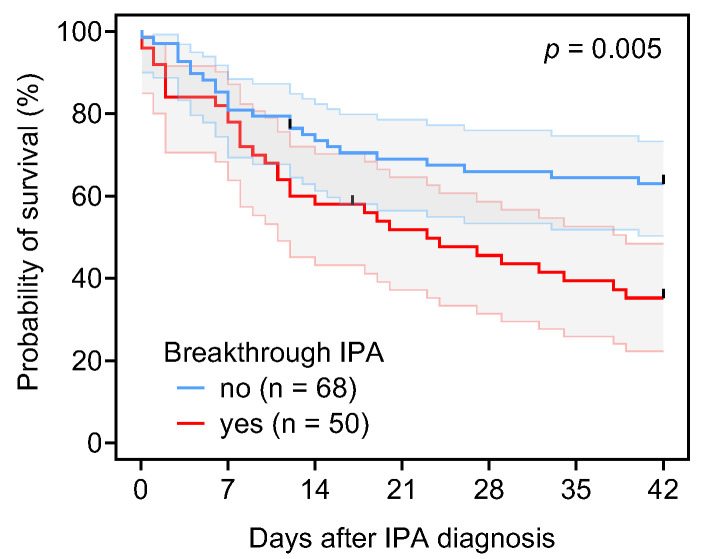
Survival curves of patients with culture-positive proven or probable IPA depending on breakthrough infection status. Error bands indicate 95% confidence intervals. Mantel–Cox log-rank test.

**Figure 3 jof-11-00623-f003:**
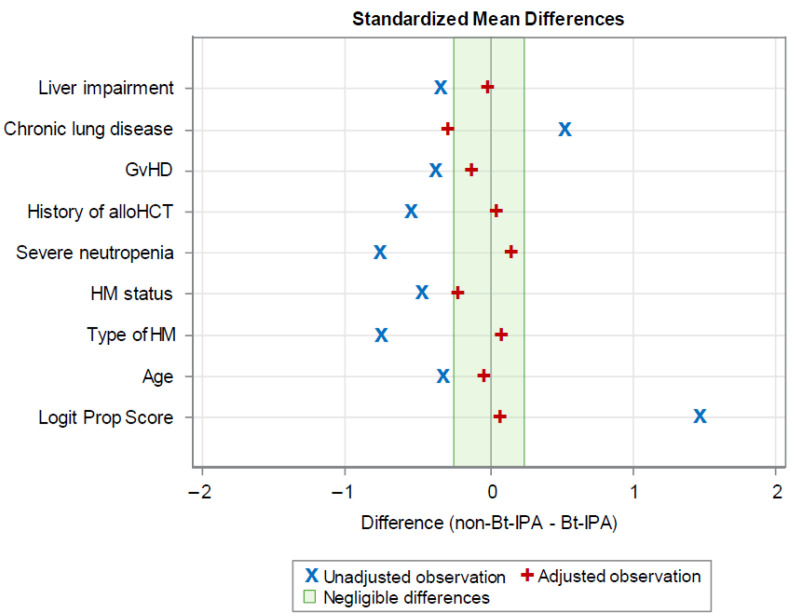
Baseline balance improvement after inverse probability of treatment weighting (IPTW) adjustment. Standardized mean differences are shown before (unadjusted) and after weighting (adjusted) for all patient characteristics included in the propensity score model. Abbreviations: GvHDs, graft-versus-host diseases; Allo-HCT, allogeneic hematopoietic cell transplantation; HMs, hematological malignancies; Logit Prop Score, Logit of propensity score.

**Table 1 jof-11-00623-t001:** Summary of clinical characteristics and comparison according to Bt-IPA status.

Characteristics ^#^	All Patients	Bt-IPA	Non-Bt-IPA	*p*-Value
	(N = 118)	(N = 50)	(N = 68)	
Baseline				
Age (years), median (range)	63 (22–89)	61 (22–77)	64 (25–89)	0.122
Gender, male, n (%)	75 (63.6)	35 (70.0)	40 (58.8)	0.213
Race, n (%)				0.359
White	81 (68.6)	32 (64.0)	49 (72.1)	
Hispanic	15 (12.7)	6 (12.0)	9 (13.2)	
Asian	10 (8.5)	5 (10.0)	5 (7.4)	
Black	8 (6.8)	6 (12.0)	2 (2.9)	
Others	4 (3.4)	1 (2.0)	3 (4.4)	
Type of hematological malignancy, n (%)				<0.001
AML/ALL/MDS	56 (47.5)	34 (68.0)	22 (32.4)	
Lymphoma/MM/CLL	62 (52.5)	16 (32.0)	46 (67.7)	
Hematological malignancy status, n (%)				0.017
Active	94 (79.7)	45 (90.0)	49 (72.1)	
Remission	24 (20.3)	5 (10.0)	19 (27.9)	
Neutropenia (ANC <500/mm^3^), n (%)	49 (41.5)	29 (58.0)	20 (29.4)	0.002
Severe neutropenia (ANC <100 mm^3^), n (%)	38 (32.2)	26 (52.0)	12 (17.7)	<0.001
History of allogenic HCT, n (%)	23 (19.5)	16 (32.0)	7 (10.3)	0.003
History of autologous HCT, n (%)	14 (11.9)	2 (4.0)	12 (17.6)	0.063
GvHD, n (%)	13 (11.0)	9 (18.0)	4 (5.9)	0.038
Hypoalbuminemia <3.0 mg/dL	54 (45.8)	25 (50.0)	29 (42.6)	0.848
Other comorbidities				
Diabetes mellitus, n (%)	19 (16.1)	6 (12.0)	13 (19.1)	0.299
Chronic renal insufficiency, n (%)	16 (13.6)	6 (12.0)	10 (14.7)	0.671
Chronic lung diseases, n (%)	12 (10.2)	1 (2.0)	11 (16.2)	0.012
Liver impairment, n (%)	32 (27.1)	18 (36.0)	14 (20.6)	0.063
Serum albumin <3.0 mg/dL, n (%)	54 (45.8)	25 (50.0)	29 (42.7)	0.428
Previous corticosteroid use, n (%)	29 (24.6)	11 (22.0)	18 (26.5)	0.577
*Aspergillus* species ^a^, n (%)				0.028 *
IPA caused by four common species ^b^	99/115 (86.1)	39 (78.0)	60/65 (92.3)	
*A. fumigatus*	48/115 (41.7)	20 (40.0)	28/65 (43.1)	0.946
Non*-A. fumigatus*	51/115 (44.3)	19 (38.0)	32/65 (49.2)	0.486
IPA caused by rare species ^c^	13/112 (11.6)	10/49 (20.4)	3/63 (4.8)	
Mixed *Aspergillus* species, n (%)	5 (4.2)	2 (4.0)	3 (4.4)	>0.999
Antifungal therapy of IPA, n (%)				0.002
Azole monotherapy	63/115 (54.8)	19 (38.0)	44/65 (67.7)	
Liposomal amphotericin B monotherapy	3/115 (2.5)	0 (0)	3/65 (4.6)	
Echinocandin monotherapy	3/115 (2.5)	1 (2.0)	2/65 (3.1)	
Azole + Liposomal amphotericin B	21/115 (17.8)	15 (30.0)	6/65 (9.2)	
Azole + Echinocandin	15/115 (13.0)	7 (14.0)	8/65 (12.3)	
Liposomal amphotericin B + Echinocandin	10/115 (8.7)	8 (16.0)	2/65 (3.1)	
**Outcome**				
42-day mortality ^d^, n (%)	57/116 (49.1)	32/49 (65.3)	25/67 (37.3)	0.003

^#^ Unless indicated otherwise, variables refer to the time of IPA diagnosis. * *p*-value for the comparison of IPA caused by four common *Aspergillus* species and that was caused by rare species. ^a^ Six isolates were reported as ‘*Aspergillus* spp.’ or ‘*Aspergillus* spp. other than *A. fumigatus*’. ^b^ *A. fumigatus*, *A. flavus*, *A. terreus*, and *A. niger*. ^c^ *A. ustus/calidoustus* (n = 5), *A. versicolor* (n = 5), *A. nidulans* (n = 2), and *A. glaucus* (n = 1). ^d^ Two patients with lost follow-up within 42 days after IPA diagnosis were not included in the analysis. Abbreviations: ALL, acute lymphoblastic leukemia; AML, acute myelogenous leukemia; ANC, absolute neutrophil count; Bt-, breakthrough; CLL, chronic lymphocytic leukemia; GvHD, graft-versus-host disease; HCT, hematopoietic cell transplantation; IPA, invasive pulmonary aspergillosis; MDS, myelodysplastic syndrome; MM, multiple myeloma.

**Table 2 jof-11-00623-t002:** Comparison of 42-day survivors and deceased patients.

**Characteristics ^#^**	**Survived** **(N = 59) ^a^**	**Died** **(N = 57) ^a^**	* **p** * **-Value**
Age (years), median (range)	60 (22–82)	64 (24–85)	0.336
Gender, male, n (%)	36 (61.0)	37 (64.9)	0.664
Race, n (%)			0.669
White	37 (62.7)	42 (73.7)	
Hispanic	10 (17.0)	5 (8.8)	
Asian	6 (10.2)	4 (7.0)	
Black	4 (6.8)	4 (7.0)	
Others	2 (3.4)	2 (3.5)	
Type of hematological malignancy, n (%)			0.064
AML/ALL/MDS	23 (39.0)	32 (56.1)	
Lymphoma/MM/CLL	36 (61.0)	25 (43.9)	
Hematological malignancy status, n (%)			0.008
Active	41 (69.5)	51 (89.5)	
Remission	18 (30.5)	6 (10.5)	
Neutropenia (ANC <500/mm^3^), n (%)	16 (27.1)	33 (57.9)	<0.001
Severe neutropenia (ANC <100/mm^3^), n (%)	11 (18.6)	27 (47.4)	0.001
Neutropenia status until day 42 after IPA diagnosis, n (%)			0.002
Neutropenia at IPA diagnosis, no recovery	10 (17.0)	26 (45.6)	
Neutropenia at IPA diagnosis, recovery	6 (10.2)	7 (12.3)	
No neutropenia at IPA diagnosis	43 (72.9)	24 (42.1)	
History of alloHCT, n (%)	11 (18.6)	12 (21.1)	0.745
GvHD, n (%)	8 (13.6)	5 (8.8)	0.414
Serum albumin <3.0 mg/dL, n (%)	16 (27.1)	37 (64.9)	<0.001
Previous steroids use, n (%)	18 (30.5)	11 (19.3)	0.163
*Aspergillus* species, n (%)			0.305
IPA caused by four common species ^b^	52/57 (91.2)	45/53 (84.9)	
IPA caused by rare species	5/57 (8.8)	8/53 (15.1)	
Mixed *Aspergillus* species, n (%)	1 (1.7)	4 (7.0)	0.203
Breakthrough IPA, n (%)	17 (28.8)	32 (56.1)	0.003
Duration of antifungal exposure before breakthrough-IPA	35 (5–90)	31 (7–83)	0.602
Antifungal therapy of IPA, n (%)			0.054
Monotherapy	40 (67.8)	27/54 ^c^ (50.0)	
Combination therapy	19 (32.2)	27/54 ^c^ (50.0)	

^#^ Unless indicated otherwise, variables refer to the time of IPA diagnosis. ^a^ Two patients with lost follow-up within 42 days after IPA diagnosis were not included in this analysis. ^b^ *A. fumigatus*, *A. flavus*, *A. terreus*, and *A. niger*. ^c^ Three patients died without receiving antifungal therapy. Abbreviations: alloHCT, allogeneic hematopoietic cell transplantation; ALL, acute lymphoblastic leukemia; AML, acute myelogenous leukemia; ANC, absolute neutrophil count; CLL, chronic lymphocytic leukemia; GvHD, graft-versus-host disease; MDS, myelodysplastic syndrome; IPA, invasive pulmonary aspergillosis; MM, multiple myeloma.

**Table 3 jof-11-00623-t003:** Independent risk factors for 42-day mortality and prognostic impact of breakthrough IPA by multivariable Cox regression analysis.

**(A) Before IPTW Adjustment**			
**Variables ^#^**	**aHR**	**95% CI**	* **p** * **-Value**
Neutropenia (ANC <500/mm^3^)	1.95	1.11–3.41	0.020
Serum albumin <3.0 mg/dL	2.52	1.42–4.47	0.002
Breakthrough IPA	1.66	0.97–2.83	0.064
**(B) After IPTW adjustment ^a^**			
**Variables ^#^**	**aHR**	**95% CI**	* **p** * **-value**
Neutropenia (ANC <500/mm^3^)	2.10	1.13–3.88	0.018
Serum albumin <3.0 mg/dL	1.84	1.00–3.40	0.052
Breakthrough IPA	1.49	0.81–2.72	0.201

^#^ at diagnosis of IPA. ^a^ The model was also adjusted for: (i) chronic lung diseases, the variable with remaining imbalance after IPTW adjustment (standardized difference >0.25), and (ii) *Aspergillus* species as a potential modulating variable that was significantly associated with Bt-IPA status in Table 1. Abbreviations: aHR, adjusted hazard ratio; ANC, absolute neutrophil count; IPA, invasive pulmonary aspergillosis; IPTW, inverse probability of treatment weighting; 95% CI, 95% Confidence Interval.

## Data Availability

Data are not publicly available due to ethical restrictions and patient confidentiality.

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
