# Peer review of "What Is the Prognostic Significance of Culture-Documented Breakthrough Invasive Pulmonary Aspergillosis in Patients with Hematological Malignancies? A Propensity Score-Adjusted Analysis"

_jof, 2025, doi:10.3390/jof11090623_

Round 1

Reviewer 1 Report

The manuscript is interesting and well written. A few suggestions to improve the MS have been provided.

See attached file. 

Author Response

Reviewer #1
The authors present a single-center retrospective cohort study analyzing culture-positive invasive pulmonary aspergillosis (IPA), comparing breakthrough (Bt) and non-breakthrough episodes. Independent predictors for 42-day all-cause mortality were assessed in the overall cohort using both unadjusted and propensity score-adjusted models. The manuscript provides valuable data in this complex and understudied area. 
Below are specific comments for the authors’ consideration:
Materials and methods:
- Defining the day of IPA diagnosis is always challenging in this patient population. Should it be the day of symptom onset, the date of the diagnostic CT scan, or the date of microbiological confirmation? This distinction is important, particularly when classifying episodes as breakthrough, as the timing of antifungal exposure may shift accordingly.
We appreciate the reviewer’s insightful comment regarding the complexity of defining the date of IPA diagnosis in this population. This issue is particularly critical in studies evaluating breakthrough infections, as the timing of antifungal exposure relative to disease onset can influence classification and interpretation. We agree that determining the precise date of diagnosis requires careful consideration. 
In practice, there is often a time gap between the onset of symptoms (which may be as non-specific as fever alone), the appearance of radiological findings suggestive of pneumonia, and the eventual microbiological confirmation. However, to our knowledge, there is no universally accepted standard regarding the maximal allowable interval between these elements to still be considered part of a single IPA episode. This inherent ambiguity is a well-known challenge in clinical studies of IPA. Therefore, for the purpose of consistency, we defined the date of diagnosis for proven/probable IPA as the day on which both microbiological evidence and compatible clinical features (symptoms and CT findings) were present. In culture-proven cases, this corresponded to the day when microbiological confirmation was obtained, with concurrent clinical and radiological manifestations. 
This definition has now been explicitly added to the Methods section (Line 81-83) as follows: 
“The date of IPA diagnosis was defined as the day of microbiological confirmation was obtained (i.e., date of specimen collection for culture confirmation) with concomitant presence of clinical findings (including CT imaging and symptoms) compatible with IPA.”

Results: 
- Figure 1: This is a clear and informative flowchart. Congratulations to the authors. A few minor suggestions:
According to Figure 1B, 17 out of 50 (34%) patients received an antifungal combination prior to the Bt episode. This information could be better presented in Table 1 for easier reference.
Thank you for your helpful suggestion. We agree that presenting the proportion of patients who received multiple antifungal agents could enhance clarity. However, as this information applies only to the Bt-IPA group, incorporating it into Table 1—which is designed to allow direct comparison between Bt and non-Bt groups—would affect the table’s structural consistency and further expand its already considerable size.
To improve accessibility of this information without compromising the table format, we have instead revised Figure 1B to indicate the number and percentage of patients who received two or more antifungal exposure prior to the Bt-IPA diagnosis more clearly. (Page 3, Figure 1B) (Figure legends, Line 94-95) 

In the figure caption, the authors state that 17 patients were exposed to more than one antifungal “concomitantly or sequentially.” Referring to the cited definition of Bt-IPA, I understand this includes patients who developed the breakthrough episode while receiving one drug, having recently received another antifungal with a potentially long half-life. This might explain, for example, why a patient received three different triazoles. However, this nuance may not be immediately clear to all readers. I recommend clarifying this in the Methods section, including how long after discontinuation of a drug an episode was still considered a breakthrough, for each antifungal agent.
Thank you for this insightful comment. We agree that the phrase “concomitant or sequential antifungal exposure” may be ambiguous and warrants further clarification.
According to the cited reference, we applied a strict case definition of Bt-IPA. Among cases that met this definition, we systematically captured all antifungal exposures within 90 days prior to IPA diagnosis, as described in Lines 68–69. Specifically, if an agent had been administered for a meaningful duration sufficient to reach steady state and any type of antifungal exposure continued uninterrupted up to the time of Bt-IPA diagnosis—whether given concomitantly or in sequence—we documented all such agents and included them in the results. However, recognizing that this could potentially cause misunderstanding, we considered “exposure to” rather than “breakthrough to” to be the more appropriate term and revised the wording. 
We have also revised the terminology in Figure 1B and its legend to improve clarity (page 3, Figure 1B). 

- Table 1:
The data on rare Aspergillus species is of interest. It would be helpful if the authors could further specify how many of the four most common species were A. fumigatus and how many belonged to the other three. 
Thank you for pointing this out. We agree that further clarification of species distribution is helpful. We have now specified the number of A. fumigatus and non-fumigatus species among the four most common isolates in Table 1. (Page 5)

The finding that lymphoma/MM/CLL accounted for over 50% of cases despite not traditionally being high-risk groups for IPA is important. Have the authors any data on BsAbs and CAR T-cell therapy use? 
We appreciate the reviewer’s insightful observation. Indeed, the high proportion of IPA cases occurring in patients with lymphoma/MM/CLL is a notable finding and may reflect evolving risk factors in this population. In our cohort, no patients had received bispecific antibody (BsAb) therapy, including blinatumonab. CAR T-cell therapy had been administered in 4 patients, of whom only 1 experienced a Bt-IPA episode. Given the small number of cases, we believe it would be premature to draw conclusions regarding the contribution of CAR T-cell or BsAb therapy to Bt-risk in our cohort.
As these therapies are increasingly used, we agree that further studies specifically designed to evaluate their impact on the epidemiology and risk of invasive mold infections are needed.

For the Bt-IPA cases, it would be very informative to know which antifungals patients received based on prior antifungal exposure (i.e., the agent against which the infection broke through). A Sankey diagram or a similar visualization could be a useful and intuitive way to present this. If not feasible, even a simple bar chart with antifungal pairings (prior vs. treatment) could help. Were rare Aspergillus species mostly related to a specific antifungal type (e.g., azoles?). 
Thank you for this valuable suggestion. We agree that visualizing prior antifungal exposure alongside subsequent treatment choices could enhance the interpretability of our data. In our cohort, we observed a general tendency to either switch antifungal class or escalate therapy following breakthrough infection—for example, the addition of liposomal amphotericin B or an echinocandin in patients who had received posaconazole prophylaxis, or switching from caspofungin to a mold-active azole for definitive treatment.
However, given the tremendous heterogeneity of clinical decisions, individualized treatment strategies, and the relatively small sample size of Bt-IPA cases (n=50), we were unable to identify consistent or statistically meaningful patterns between prior and treatment antifungal agents. Likewise, the study was not powered to discern a clear association between rare Aspergillus species and specific antifungal exposures (e.g., azoles). 
For review purposes, we have generated the following exploratory chart to illustrate antifungal transitions in Bt-IPA cases: 

However, given that no individual combination of breakthrough exposures and subsequent therapy was seen in more than 3 patients, we believe that this visualization does not imply any consistent or generalizable relationship between the breakthrough agent and subsequent definitive therapy. The observed treatment patterns reflect retrospective, expert-driven decisions at a single center rather than standardized protocols. Furthermore, the use of combination therapy is already described in Table 1. For these reasons, we have opted not to include the figure in the manuscript. 

- Section 3.5:
Please consider revising the sentence:
â–ª “Serum TDM results within 5 days of Bt-IPA diagnosis were available for most patients (23/30, 76.7%) who were exposed to voriconazole (10/14, 71.4%) or posaconazole (13 out of 16, 81.3%)”
• To: “Serum TDM results within 5 days of Bt-IPA diagnosis were available for most patients (23/30, 76.7%), including 10/14 (71.4%) exposed to voriconazole and 13/16 (81.3%) exposed to posaconazole.”
We appreciate the reviewer’s suggestion and have revised the sentence accordingly (Line 188-190). 

- Section 3.6:
I am not certain about journal limits for figures, but if space allows, I recommend presenting Figure 1D as a separate figure. The mortality differences are of high interest, and the current layout may reduce their visibility. 
Thank you for the suggestion. Given the clinical importance of mortality outcomes, we have now presented Figure 1D as a separate figure as Figure 2 (Page 7, Line 200-205).

The definition of hypoalbuminemia should be provided in the Methods section. This finding is notable—was albumin data available for all patients?
We thank the reviewer for this comment. Hypoalbuminemia was defined as serum albumin <3.0 g/dL, as already noted in Tables 1 and 2. We have now added this definition to the manuscript for clarity (Line 208).

Did any patient receive immunotherapy (e.g., granulocyte transfusions, immune checkpoint inhibitors)?  
Thank you for this important question. Our study was not originally designed to collect detailed data on immunotherapy; however, in our cohort, we identified three patients who had received the immune checkpoint inhibitor nivolumab—one with chronic lymphocytic leukemia (CLL) and two with lymphoma. All three had active underlying hematologic diseases at the time of IPA diagnosis. Only one of these patients developed Bt-IPA, and this patient subsequently died.
Granulocyte transfusions was not included as a study variable. At MDACC, it is not used as a main supportive therapeutic modality, and its clinical benefit is considered limited. Given the short half-life of transfused granulocytes, we believe it is unlikely that granulocyte transfusion would have had a meaningful impact on the occurrence of Bt-IPA or on IPA outcomes in our study population.
Given the small number of cases received immune checkpoint inhibitor, we did not perform further analysis, but we agree that immunotherapy may represent a relevant factor for future investigations.

As expected in this context, mortality was high. Do the authors have data on the cause of death? Was mortality directly attributable to IPA?  
We appreciate this important question. In this study, mortality was assessed as all-cause mortality due to the retrospective design. Attribution to IPA was not systematically determined due to the retrospective design. However, we know from an ongoing study (to be presented at ID Week 2025) that >90% of 6-week mortality in patients with hematologic malignancy and IPA at our institution are either IPA-attributable or at least IPA-contributable. 

Discussion:
- One of the most complex and intriguing aspects of Bt-IFI is understanding why antifungal therapy fails. Although the authors rightly point out that there is no definitive correlation between in vitro susceptibility and clinical outcomes, MIC data for the antifungal agents used prior to breakthrough would have added valuable insight into potential failure causes.
We fully agree that MIC data would have strengthened the interpretation of antifungal treatment outcomes. Unfortunately, MIC testing was not routinely performed during the study period at our institution. This limitation has been acknowledged in the manuscript: “Thirdly, we did not assess antifungal resistance or cryptic Aspergillus species identification, as these are not routinely performed in our microbiology laboratory and data for in-vitro/in-vivo correlation of antifungal susceptibilities and clinical outcomes of IA are still lacking [33].” (Line 329-332).

- While the authors acknowledge that cryptic species identification was not performed, it is unclear how species were identified. The microbiological methodology appears to be missing or underdeveloped in the Methods section and should be expanded.
Cryptic species identification would require molecular sequencing, which is not routinely performed in most clinical microbiology laboratories, including ours. We have expanded the description of the microbiological identification process in the Methods section as follows.

“Information regarding Aspergillus spp. identification was derived from clinical microbiology laboratory, based. macroscopic and microscopic morphology, growth characteristics, and MALDI-TOF MS (where applicable). Molecular identification (e.g., sequencing) to detect cryptic Aspergillus species was not performed as has not been part of routine laboratory practice.” (Line 70-74) 

Reviewer 2 Report

I have reviewed the manuscript entitled “What is the prognostic significance of culture-documented breakthrough invasive pulmonary aspergillosis in patients with hematological malignancies? A propensity score-adjusted analysis.” Submitted by Sung-Yeon Cho et al.

The manuscript describes a retrospective study evaluating 118 culture-positive proven/probable cases of invasive aspergillosis and compares the 42-day all causes mortality using a propensity score-adjusted Cox regression model.

General:

The manuscript is clear and well written. The methodology used supports the authors conclusions. There are several limitations as the authors state including the inability to evaluate risk factors for developing break-through IA, it is a single center study and is a retrospective study using fungal cultures. Overall conclusion that bt-IPA was more common in high-risk patients is not unexpected, or that neutropenia was independently associated with higher 42-day mortality.

Specific:

Results:

Line 169-170: states “bt-IPA patients more commonly received combination therapy, while most non-Bt IPA received azole monotherapy”. Not clear what this means? Were bt-IPA already receiving combination antifungals at time of positive culture and breakthrough? That has not been the standard of care. Please clarify.

Figure 2: Not necessary for this paper. Results are in the text.

Detailed comments included above

Reviewer 3 Report

The authors of this manuscript wished to explore the epidemiology and associated factors for proven/probable break through invasive pulmonary aspergillosis (Bt-IPA) in a group of hematologic malignancy patients as this infection produces considerable morbidity and mortality despite the widespread use of mold-active prophylactic antifungal agents. This retrospective review was conducted in consecutive adult patients with hematologic malignancies selected on the basis of the presence of culture positive proven/probable IPA between 1/2026 and 12/2021. The primary endpoint was 42-day mortality after the first positive Aspergillus culture.

The manuscript is well written and addresses the subject matter. The Consort figure provided outlining previous antifungal exposure is very helpful. The Results section is robust and the Discussion section provides an adequate commentary on the results presented.

There are some issues which should be addressed in a revised version of the manuscript:

  1. The authors do inform the reader of how many of the patients in total had allo-HSCT but should combine this with the information on prior diagnosis of AML/ALL/MDS/lymphoma. Similarly, they should also inform readers of the number of lymphoma/MM that may have had autologous-SCT and CAR-T therapy.
  2. With reference to comment 1, these data should be described in the clinical characteristics.
  3. There is a thorough description of the TDM for the azole antifungals used within 5 days of the diagnosis of Bt-IPA.
  4. The paragraph involving lines 201-207, should also describe the appearance of neutropenia, severe neutropenia and hypoalbuminemia in the specific patient groups of Bt-IPA and non-Bt-IPA beyond the description in survivors versus non-survivors.
  5. What is meant by the sentence in lines 215-218 with regard to improved balance between the Bt-IPA and non-Bt-IPA groups. Why is achieving balance between the groups by IPTW important. Thus, the reasoning behind Figure 3 should be explained better.
  6. Greater explanation of the sentence in lines 220-221 is needed. This is not in keeping with the overall 42 day mortality that was greater in the Bt-IPA versus non-Bt -IPA groups (65.3% versus 37.3%).

These issues should be addressed in a revised manuscript.

As above.

Round 2

Reviewer 1 Report

I thank the authors for thoroughly addressing the reviewer's comments

N.A.

Reviewer 3 Report

No issues identified in revised manuscript. It is now acceptable for publication.

None.